# Chemometric Screening of Oregano Essential Oil Composition and Properties for the Identification of Specific Markers for Geographical Differentiation of Cultivated Greek Oregano

Eleftheria S. Tsoumani [ID], Ioanna S. Kosma *[ID] and Anastasia V. Badeka *

Laboratory of Food Chemistry, Department of Chemistry, University of Ioannina, GR-45110 Ioannina, Greece
* Correspondence: i.kosma@uoi.gr (I.S.K.); abadeka@uoi.gr (A.V.B.)

**Abstract:** The present study investigated the potential interconnection between the place of cultivation of Greek oregano samples and the composition and properties of their essential oils (EOs). In addition, it attempted to identify characteristic chemical features that could differentiate between geographical origins with the use of chemometric tools. To this end, a total of 142 samples of commercially available Greek oregano (*Origanum vulgare* ssp. *hirtum*) plants harvested during the calendar years 2017–2018 were obtained for this study. The samples came from five different geographical areas of Greece and represented twelve localities. After appropriate processing, the oregano samples were subjected to hydrodistillation (HD), and the resulting EOs were analyzed for their total phenolic content (TPC), antioxidant activity, and chemical composition. The acquired data were subjected to the chemometric methods of multivariate analysis of variance (MANOVA) and linear discriminant analysis (LDA) to investigate the potential of classifying the oregano samples in terms of geographical origin. In addition, stepwise LDA (SLDA) was used as a final step to narrow down the number of variables and identify those wielding the highest discriminatory power (marker compounds). Carvacrol was identified as the most abundant component in the majority of samples, with a content ranging from 28.74% to 68.79%, followed by thymol, with a content ranging from 7.39% to 35.22%. The TPC values, as well as the Trolox equivalent antioxidant capacity (TEAC) values, showed no significant variations among the samples, ranging from $74.49 \pm 3.57$ mg GAE/g EO to $89.03 \pm 4.76$ mg GAE/g EO, and from $306.83 \pm 5.01$ µmol TE/g EO to $461.32 \pm 7.27$ µmol TE/g EO, respectively. The application of the cross-validation method resulted in high correct classification rates in both geographical groups studied (93.3% and 82.7%, respectively), attesting to a strong correlation between location and oregano EO composition.

**Keywords:** Greek oregano; essential oils; geographical differentiation; chemometrics



## 1. Introduction

A trend toward healthier lifestyles has recently emerged among consumers, leading to increasing demand for herbal medicines, nutraceuticals, and natural foods worldwide. Medicinal and aromatic plants (MAPs), one of the wealthiest bioresources of drugs in traditional and modern medicine and an abundant source of fragrances, condiments, decoctions, and essential oils (EOs), have become a burgeoning area of research due to their treasured active ingredients [1]. An extensively employed herb that enjoys wide industrial, pharmaceutical, and traditional usage worldwide is oregano. Oregano, apart from its proven biological (antimicrobial, fungicidal, and antioxidant) properties, has a unique aroma that distinguishes it from other plants [2–7]. The term "oregano", attributed to more than 60 species globally, is mainly associated with the genus *Origanum* of the Lamiaceae family, which is for the most part spread throughout the Mediterranean [8]. *Origanum* presents excellent morphological and chemical diversity and is assorted into 49 taxa and 42 species. In most European countries, *Origanum vulgare* L. is the most predominant species of the genus [9,10].

Greek oregano, or *Origanum vulgare* L. ssp. *hirtum* (Link) Ietswaart, is regarded as one of the best varieties in the world in terms of quality due to its EO composition and high EO yield [11–14]. Previously published data have revealed significant variability in the chemical composition of the EOs of oregano and their yield, even within species [14–16]. The main constituents of the EOs of Greek oregano include four biosynthetically related monoterpene compounds: $\gamma$-terpinene, *p*-cymene, and either thymol or carvacrol, depending on the chemotype [8,17]. The chemotypes of aromatic plants are generally defined by the predominant compound of their EO. In the case of Greek oregano, the types that prevail are the carvacrol type; the thymol type; and the carvacrol/thymol type, wherein carvacrol and thymol are present in almost equal amounts. As a rule, the carvacrol chemotype designates a condiment as oregano; however, the amount of carvacrol may vary significantly among *O. vulgare* plants (from traces to over 90%) depending on the region, season, and subspecies [18]. While studying autumnal Greek oregano plants from several parts of Greece, Kokkini et al. (1997) [19] recorded noticeable differences in their total EO content and the concentration of their four main components: the $\gamma$-terpinene content ranged from 0.6 to 3.6% of the total EO, *p*-cymene from 17.3 to 51.3%, thymol from 0.2 to 42.8%, and carvacrol from 1.7% to 69.6%. Additionally, when comparing these data to those obtained from plants collected from the same localities in the mid-summer, the authors found that the carvacrol ratio was much higher in the summer, while in the autumn, *p*-cymene predominated. Likewise, Russo et al. (1998) [20] reported significant quantitative and quality variations when studying the chemical composition of wild populations of *Origanum vulgare* ssp. *hirtum* in Calabria, Italy.

The analysis of the active ingredients of Greek oregano EOs can be challenging due to the aforementioned chemical diversity and variability. Such challenges can be met using chemometrics, a discipline that integrates mathematics, statistics, and formal logic and provides helpful information through processing multivariate chemical data [21]. Qualitative chemometric models are widely used in food analysis to determine authentication, trace geographical or genetic origins, and detect impurities. In contrast, quantitative models are mainly used to estimate concentrations of food ingredients [22]. Chromatographic methods such as gas chromatography (GC), liquid chromatography (LC), high-performance liquid chromatography (HPLC), and high-temperature gas chromatography (HTGC) are often coupled with chemometrics to identify unique marker compounds that could indicate differentiation with respect to the place of origin [23]. Most studies using these methods have focused on honey [24–26] and dairy products [27–29], whereas fewer have included spices such as saffron [30], paprika [31], and oregano [14]. Even though studies on Greek oregano and its many properties are abundant in the literature, Vokou et al. [14] were the first, and to the best of the authors' knowledge the only, authors who attempted to correlate the chemical properties of wild *O. vulgare* ssp. *hirtum* to its geographical origin. However, rather than attempting to correlate the sampling regions to the composition and yield of the oregano EOs, they employed multifactor ANOVA to process the geographical and climatic characteristics of the areas, aiming to identify their effect on the attributes of oregano. In particular, they assessed six factors—altitude, distance from the sea, moisture index, summer water deficiency, thermal efficiency (TE), and summer concentration of TE—in relation to the EO yield; the concentration sum of thymol and carvacrol; and the concentration sum of thymol, carvacrol, $\gamma$-terpinene, and *p*-cymene. They observed that four out of the six factors (altitude, summer water deficiency, TE, and summer concentration of TE) significantly affected the yield, whereas only thermal efficiency appeared to influence the compound concentrations.

Despite the sharp increase in consumption and the significant commercial value of Greek oregano, coordinated efforts to domesticate and systematically cultivate oregano in Greece have only begun in recent decades [32–34]. Because oregano growers more often than not use oregano populations without any appropriate plant material selection, a wide array of products of varying quality, particularly in terms of composition, are commercially produced. Apart from generating inconsistency, such tactics may leave room

for acts of profiteering and fraud due to adulteration. Additionally, the individual morpho- and ontogenetic variability and the ecological and environmental effects add to the vast heterogeneity of the species [8,14,34–38]. Therefore, a comparative study of Greek oregano EOs from different regions of Greece is a valuable tool to explore the cultivated species' chemical diversity and realize their actual commercial value.

Thus, the aims of this study were: (i) to determine the main constituents of the essential oil of Greek oregano plants collected from cultivated populations from twelve different localities of Greece with the use of gas chromatography–mass spectrometry (GC-MS); (ii) to assess the total phenolic content and antioxidant activity of these essential oils; and finally (iii) to combine the acquired data with chemometric methods in an attempt to identify characteristic chemical attributes, also known as marker compounds, that potentially signify a differentiation of geographical origin. To achieve this, the acquired data were initially treated with MANOVA; the geographical origin was set as the independent variable, while the experimental data were appointed as the dependent variables. After establishing the significant dependent variables for geographical differentiation, LDA was then applied to these designated variables in order to explore the possibility of classifying the oregano samples according to their geographical origin. The combination of multiple analytical parameters resulted in a greater aggregation of the oregano samples in the respective regions. Since the number of significant variables ($p < 0.05$) resulting from MANOVA was quite large, stepwise LDA (SLDA) was subsequently employed so as to reduce the parameters to those considered as the best set of authenticity predictors/markers in relation to the herein-studied regions.

This study is the first to focus exclusively on cultivated Greek oregano samples from areas throughout Greece. In addition, the combination of analytical parameters with the specific cultivation origin of the Greek oregano samples constitutes the novelty of the present work.

## 2. Materials and Methods

### 2.1. Oregano Samples

A total of 142 Greek oregano samples were obtained at the stage of optimum maturity from professional oregano growers in late June–early July of 2017 and 2018, respectively. The plants originated from five geographical areas of Greece (Epirus, Thessaly, Northern Greece, the Peloponnese, and Crete) and twelve localities in total (Figure 1). The samples, consisting of dried oregano aerial parts, were all processed by hand to ascertain homogeneity. The resulting rubbed oregano samples were refrigerated within air-tight glass containers until the analyses.

All oregano growers procured the initial oregano seedlings from the Hellenic Agricultural Organization, Elgo-Dimitra, as certified O. vulgare ssp. hirtum material. Moreover, the selected oregano plants were of similar cultivation characteristics: organic, open-field, mostly non-irrigated, and beyond their second year of cultivation.

### 2.2. Extraction of EOs

The EOs were extracted from the rubbed oregano samples using hydrodistillation (HD), a method chosen because it is devoid of organic solvents and is extensively used in the food industry. Exactly 20 g of each oregano sample and 300 mL of distilled water were placed into a 500 mL round-bottomed flask, which was then connected to a Clevenger apparatus (Auxilab, Spain). A heating mantle was used as a heating medium, and the samples were subjected to HD for 3 h [39]. After being dried over anhydrous sodium sulfate, the obtained EOs were filtered and stored in 3 mL amber glass bottles at 4 °C until use.

Northern Greece
(Central & Western Macedonia)
1.  Thessaloniki
2.  Katerini
3.  Kilkis
4.  Kozani

Thessaly
5.  Volos
6.  Kalambaka

Epirus
7.  Ioannina
8.  Preveza

Peloponnese
9.  Ileia
10. Achaea

Crete
11. Rethymno
12. Heraklion

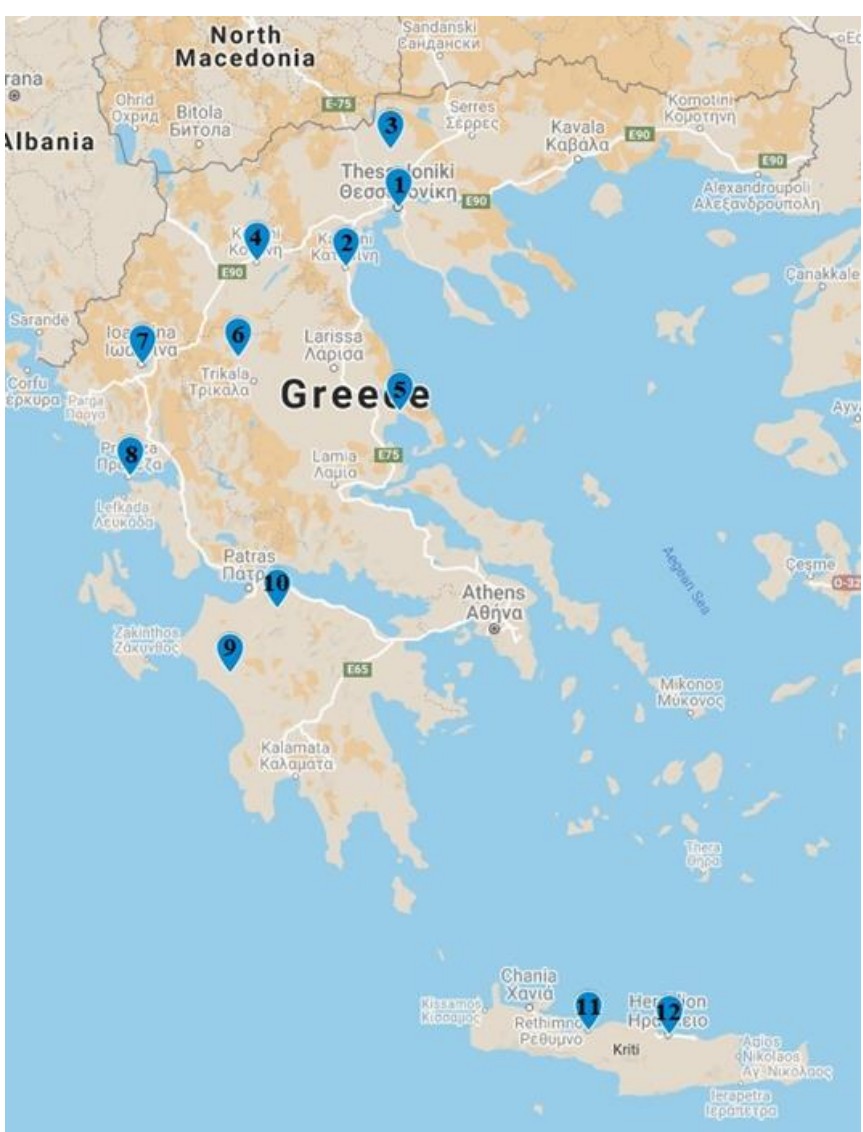

**Figure 1.** Oregano cultivation and sampling sites.

*2.3. Gas Chromatography–Mass Spectrometry (GC/MS) Instrumentation and Analysis Conditions*

The GC detector used in this study was an Agilent 7890A coupled with an Agilent 5975C inert XL MSD mass selective detector (Agilent, Wilmington, DE, USA). The GC fused silica capillary column was a BP20 (WAX) polar column, 25 m × 0.32 mm × 0.25 mm (J & W Scientific, Folsom, CA, USA.), and the carrier gas used was ultra-high purity helium at a flow rate of 1.5 mL/min. The injector operated in split mode (30:1 split ratio), and its temperature was kept at 260 °C.

An aliquot of 50 μL of each essential oil sample and 200 μL of a 4-methyl-2-pentanol internal standard solution were placed in a 5 mL volumetric flask. The flask was filled with hexane to the mark, and 1 μL of the final solution was then injected into the GC inlet port.

The temperature program used was as follows: The oven temperature was initially maintained at 40 °C for 4 min, increased to 120 °C at a rate of 20 °C/min, maintained for 2 min, raised again to 200 °C at 8 °C/min, and increased anew to 230 °C at a rate of 15 °C/min. The final temperature was maintained for 1 min, while a solvent delay was also set at 1.5 min. The acquisition was performed in the MS, operating with electron impact ionization (EI, 200 eV) and 2.92 scans/s in a 35–300 (*m/z*) mass range, while the transfer line temperature was set at 230 °C. Peak identification was performed by comparing the eluting compounds' retention times and mass spectra to the Wiley Library database [40].

Retention indices (RI) of the EO compounds were calculated using appropriate n-alkane (C8–C20) standards (Fluka, Buchs, Switzerland). All determinations were conducted thrice.

### 2.4. Determination of Total Phenolic Content (TPC) and Antioxidant Activity

The process followed regarding the methanolic extracts was based on liquid–liquid extractions and, more specifically, olive oil assays [41,42]. Precisely 0.1 g of each EO sample was mixed with 2 mL of hexane and 3 mL of MeOH/$H_2O$ (60:40). The mixture was initially vortexed for 2 min and then centrifuged at 4000 rpm for 10 min at 4 °C to ascertain the separation of the two phases. The methanol phase was separated, and the process was repeated. The combined methanolic extracts were collected in a 10 mL volumetric flask, which was then filled with MeOH/$H_2O$ (60:40) up to the mark. This methanolic sample solution was used in the following determinations.

TPC was spectrophotometrically estimated according to the Folin–Ciocalteu colorimetric method [43]. The reaction mixture was prepared in a 100 mL volumetric flask by mixing 0.2 mL of the methanolic sample solution, 0.25 mL of Folin–Ciocalteu reagent, and 2.3 mL of $H_2O$. After 3 min, 0.5 mL of $Na_2CO_3$ 20% was added to the mixture, which was then supplemented with water up to the mark. The samples were incubated at room temperature for 30 min in the dark, and the absorbance was measured at λmax = 725 nm against a blank using a Perkin Elmer Lambda 25 UV/VIS Spectrophotometer. The TPC concentrations were estimated using a calibration curve obtained over the range of 50–200 mg/kg of gallic acid. The results were expressed as mg of gallic acid equivalents (GAE)/g of oregano EO.

The antioxidant activity of the samples was assessed according to the DPPH (2,2-diphenyl-1-picrylhydrazyl) free-radical scavenging method [44,45]: 2.9 mL of DPPH solution was mixed with 0.1 mL of the methanolic sample solution and kept at room temperature for 30 min in the dark. The control solution consisted of methanol and DPPH, and the absorbance was measured at 517 nm. A calibration curve was created in the range of 10–175 mg/kg of Trolox, and the DPPH radical scavenging activity was expressed as μmol of Trolox equivalents per EO g (μmol TE/g EO sample).

All determinations were carried out in triplicate, and the results are presented as the mean average.

### 2.5. Statistical Analysis

IBM SPSS 25.0 [46] was used for all statistical analyses in this study. The acquired data were subjected to MANOVA to determine the significant variables for the geographical differentiation of oregano. Geographical origin was set as the independent variable, while several analytical parameters were selected as the dependent variables (essential oil composition, total phenolics, antioxidant capacity, and combinations thereof). After that, linear discriminant analysis (LDA) was applied using the same parameters to identify characteristic chemical attributes that potentially differentiated between geographical origins. The original and leave-one-out cross-validation methods were implemented to evaluate the prediction classification ability. The procedure was repeated for all the parameters of the samples. Box's M test was conducted to assess the homogeneity of variability in this study [47,48].

Finally, stepwise LDA (SLDA) was applied as the ultimate classification method to distinguish the most significant variables through a stepwise process in order to optimize the discrimination. The classification evaluation of SLDA was conducted by leave-one-out cross-validation [49,50].

## 3. Results and Discussion

### 3.1. Essential Oil Chemical Composition

The composition of the essential oil of each oregano sample, as determined by GC and combined GC-MS, is shown in Table 1. In total, 35 compounds and two chemotypes were identified. The following fifteen compounds were present in all samples: *α*-pinene, *β*-myrcene, *α*-terpinene, *γ*-terpinene, *p*-cymene, 1-octen-3-ol, *cis*-sabinene hy-

drate, *trans*-sabinene hydrate, caryophyllene, 4-terpineol, borneol, *β*-bisabolene, caryophyllene oxide, thymol, and carvacrol. However, these were present in varying proportions, and carvacrol was the predominant constituent in all samples apart from the Ileia oregano. The fundamental components of the EOs were primarily oxygenated monoterpenes (65.67–83.98%) and monoterpene hydrocarbons (10.80–30.43%). Sesquiterpene hydrocarbons, oxygenated sesquiterpenes, and miscellaneous compounds followed at lower rates: 2.44–3.97%, 0.40–1.92%, and 0.10–1.29%, respectively.

The most abundant regions in terms of compounds were Kozani and Ioannina, each recording 33 in total, followed closely by Preveza, Ileia, and Achaea with 31, 28, and 27 compounds, respectively. Contrarily, the region of Thessaloniki presented the fewest constituents, featuring only 16 in total, while the rest of the studied locations recorded around 22 compounds each.

Most oregano samples pertained to the carvacrol chemotype, with carvacrol rates ranging from 46.19% (Rethymno) to 68.79% (Thessaloniki). In contrast, only two belonged to the carvacrol/thymol chemotype, with both sampling regions situated in the Peloponnese: Ileia, with 28.74% carvacrol and 35.22% thymol, and Achaea, with 34.77% carvacrol and 34.68% thymol. This finding contradicted the results previously reported by Vokou et al. [14], who found that the oregano essential oil samples from the Peloponnese were primarily composed of carvacrol. In a similar study on *Origanum vulgare* L. subsp. *hirtum* cultivated in Turkey, Esen et al. [39] reported contents of carvacrol and thymol that varied significantly from 5.3 to 85.4% and from 0.3 to 68.0%, respectively. Nevertheless, these diverging results further support the assertion that the species boasts considerable variability, so such fluctuations are to be expected [3,14].

Similar to previous reports, the four principal components present in considerable amounts in all samples were the aromatic monoterpenes carvacrol, thymol, *p*-cymene, and *γ*-terpinene [15,17,19,39,51–56]. Despite the quantitative variations in these main components, their sum content appeared almost equivalent in the EOs of different regions and represented more than 80% of the total oil, specifically ranging between 85.12% and 89.51%. These results aligned with the findings of past studies. In particular, Kokkini et al. and Vokou et al. [3,14] reported a similar range (85.0 to 96.8%) when studying the EO composition of *O. vulgare* ssp. *hirtum* of different geographic origins as well as over different seasons. Thymol, the second most abundant ingredient, recorded its highest values in the oregano of Ileia and Achaea (35.22% and 34.68%, respectively), both located in the Peloponnese, and its lowest in the Heraklion sample (7.39%). Additionally, the lowest percentages of *γ*-terpinene and *p*-cymene were recorded in the Thessaloniki sample (2.72% and 6.60%, respectively), whereas the highest rates were registered for *γ*-terpinene in the Ileia samples (10.49%) and *p*-cymene in the Heraklion samples (12.53%).

Delving deeper into the EO composition, six additional compounds—four monoterpenes (*β*-myrcene, *α*-terpinene, 4-terpineol, and borneol) and two sesquiterpenes (caryophyllene and *β*-bisabolene)—exhibited relatively high content rates in all samples. Caryophyllene, a common bicyclic sesquiterpene, was the fifth most abundant constituent in 8 out of the 12 regions examined, with contents ranging from 1.72% (Ioannina) to 2.43% (Katerini). As for the remaining four regions, the fifth most abundant component was the monoterpene *α*-terpinene in the Ileia and Achaea samples (2.48% and 1.63%, respectively); *β*-myrcene in the Rethymno samples (1.65%); and the sesquiterpene *β*-bisabolene in the Heraklion samples (2.07%). In addition to these compounds, the oxygenated monoterpenes 4-terpineol and borneol were also notably present in all samples. The lowest content of 4-terpineol was recorded in the sample originating from the region of Ileia (0.17%), and the highest was registered in the sample from Kozani (1.28%). Likewise, borneol reached its highest content rate in the oregano sample from Thessaloniki (1.41%) and its lowest in the samples from Ileia and Rethymno (both 0.51%).

**Table 1.** Essential oil chemical composition (%) for each of the twelve regions studied. Main compounds and their values are marked in bold blue.

| S/N [a] | Compounds | RI$_{EXP}$ | RI$_{LIT}$ | THESSALONIKI | KATERINI | KILKIS | KOZANI | VOLOS | KALAMBAKA | IOANNINA | PREVEZA | ILEIA | ACHAEA | RETHYMNO | HERAKLION |
|---|---|---|---|---|---|---|---|---|---|---|---|---|---|---|---|
| 1 | α-pinene | 1012 | 1012 | 0.26 | 0.62 | 0.60 | 0.75 | 0.53 | 0.85 | 0.62 | 0.73 | 0.90 | 0.54 | 0.71 | 0.80 |
| 2 | α-thujene | 1019 | 1017 | | 0.09 | 0.56 | 0.90 | 0.45 | 0.59 | 0.70 | 0.96 | 1.03 | 0.68 | 0.67 | 0.06 |
| 3 | camphene | 1054 | 1055 | | | | 0.16 | tr [b] | | 0.13 | 0.12 | 0.16 | 0.07 | | |
| 4 | β-pinene | 1096 | 1100 | | | | 0.15 | | | 0.11 | 0.12 | 0.17 | 0.07 | 0.06 | |
| 5 | δ-3-carene | 1143 | 1138 | | | | | | | 0.06 | | 0.09 | | | |
| 6 | α-phellandrene | 1166 | 1166 | | | | 0.16 | 0.19 | 0.05 | 0.18 | 0.22 | 0.29 | 0.22 | | 0.07 |
| 7 | β-myrcene | 1175 | 1175 | 0.55 | 0.91 | 1.39 | 1.46 | 1.20 | 1.32 | 1.51 | 1.51 | 2.01 | 1.47 | 1.65 | 1.15 |
| 8 | α-terpinene | 1191 | 1201 | 0.67 | 1.09 | 1.45 | 1.63 | 1.22 | 1.57 | 1.46 | 1.58 | 2.48 | 1.63 | 1.56 | 1.26 |
| 9 | limonene | 1199 | 1208 | | 0.14 | 0.16 | 0.25 | 0.12 | 0.22 | 0.24 | 0.26 | 0.38 | 0.29 | 0.29 | 0.19 |
| 10 | β-phellandrene | 1221 | 1228 | | 0.10 | 0.18 | 0.29 | 0.08 | 0.11 | 0.28 | 0.28 | 0.29 | 0.28 | 0.24 | 0.07 |
| 11 | *γ*-terpinene | 1243 | 1246 | **2.72** | **3.13** | **5.76** | **7.05** | **5.40** | **6.67** | **6.96** | **7.00** | **10.49** | **8.18** | **7.28** | **4.13** |
| 12 | 3-octanone | 1255 | 1252 | | | | 0.09 | 0.21 | 0.14 | 0.22 | 0.24 | 0.23 | 0.24 | | |
| 13 | *p*-cymene | 1270 | 1269 | **6.60** | **11.10** | **8.32** | **9.77** | **8.93** | **12.26** | **8.47** | **10.45** | **11.98** | **9.98** | **12.03** | **12.53** |
| 14 | α-terpinolene | 1280 | 1282 | | | | tr | 0.14 | tr | 0.15 | 0.15 | 0.16 | 0.12 | 0.11 | tr |
| 15 | 1-octen-3-ol | 1448 | 1447 | 0.41 | 0.57 | 0.50 | 0.34 | 0.52 | 1.07 | 0.54 | 0.45 | 0.32 | 0.41 | 0.10 | 0.43 |
| 16 | *cis*-sabinene hydrate | 1462 | 1471 | 0.49 | 0.51 | 0.67 | 0.58 | 0.56 | 0.36 | 0.50 | 0.55 | 0.49 | 0.68 | 0.43 | 0.52 |
| 17 | *trans*-sabinene hydrate | 1546 | 1556 | 0.39 | 0.43 | 0.50 | 0.46 | 0.50 | 0.44 | 0.37 | 0.39 | 0.34 | 0.47 | 0.38 | 0.54 |
| 18 | bornyl acetate | 1561 | 1566 | | | | | | | tr | | | | | |
| 19 | caryophyllene | 1598 | 1590 | 1.75 | 2.43 | 2.08 | 1.92 | 1.77 | 1.96 | 1.72 | 1.74 | 1.50 | 1.13 | 0.74 | 1.45 |
| 20 | 4-terpineol | 1604 | 1605 | 1.22 | 0.85 | 0.82 | 1.28 | 1.20 | 0.33 | 1.00 | 0.88 | 0.17 | 0.89 | 0.80 | 1.25 |
| 21 | carvacrol methyl ether | 1610 | 1601 | 0.79 | 0.88 | 0.61 | 0.43 | 0.41 | 0.64 | 0.20 | 0.27 | | 0.27 | | 0.70 |
| 22 | *cis*-dihydrocarvone | 1629 | - | | | | 0.05 | | | 0.16 | 0.05 | | | | |
| 23 | α-humulene | 1668 | 1677 | | 0.09 | 0.10 | 0.27 | 0.05 | | 0.23 | 0.21 | 0.10 | 0.09 | | |
| 24 | α-terpineol | 1698 | 1698 | | | | tr | | | | | | | | |
| 25 | borneol | 1702 | 1717 | 1.41 | 1.16 | 1.06 | 0.90 | 1.20 | 1.03 | 0.80 | 0.75 | 0.51 | 0.65 | 0.51 | 0.84 |
| 26 | β-bisabolene | 1726 | 1722 | 1.33 | 1.45 | 0.87 | 1.24 | 1.19 | 1.53 | 1.06 | 1.10 | 1.32 | 1.22 | 1.45 | 2.07 |
| 27 | δ-cadinene | 1733 | 1756 | | | | 0.25 | | | 0.13 | 0.17 | | | | |
| 28 | *p*-cymen-8-ol | 1851 | 1865 | | | | 0.06 | | | | tr | | | | tr |
| 29 | carvacryl acetate | 1875 | 1880 | | | | tr | | | tr | | | | | |
| 30 | caryophyllene oxide | 1973 | 1994 | 1.72 | 1.31 | 0.96 | 0.68 | 1.00 | 0.72 | 0.66 | 0.68 | 0.37 | 0.52 | 0.40 | 0.98 |
| 31 | spathulenol | 2122 | 2136 | | | | 0.08 | | | tr | tr | 0.06 | | | |
| 32 | 4-isopropyl-m-cresol | 2155 | - | | | | 0.08 | | | tr | 0.14 | 0.20 | 0.17 | 0.13 | |
| 33 | thymol | 2173 | 2186 | **10.89** | **10.22** | **12.85** | **8.41** | **13.03** | **16.45** | **10.64** | **13.65** | **35.22** | **34.68** | **24.01** | **7.39** |
| 34 | 5-isopropyl-m-cresol | 2208 | - | | | | 0.10 | tr | | 0.10 | 0.18 | | 0.26 | 0.14 | |
| 35 | carvacrol | 2211 | 2212 | **68.79** | **62.95** | **60.28** | **59.89** | **60.04** | **51.62** | **60.05** | **55.09** | **28.74** | **34.77** | **46.19** | **63.55** |
| | **Main compounds [c]** | | | **89.00** | **87.40** | **87.21** | **85.12** | **87.40** | **87.00** | **86.12** | **86.19** | **86.43** | **87.61** | **89.51** | **87.60** |
| | Monoterpene hydrocarbons | | | 10.80 | 17.18 | 18.58 | 22.74 | 17.98 | 23.59 | 20.87 | 23.38 | 30.43 | 23.53 | 24.60 | 20.26 |
| | Oxygenated Monoterpenes | | | 83.98 | 77.00 | 76.79 | 72.24 | 76.94 | 70.87 | 73.82 | 71.95 | 65.67 | 72.84 | 72.59 | 74.79 |
| | Sesquiterpene hydrocarbons | | | 3.08 | 3.97 | 3.05 | 3.68 | 3.01 | 3.49 | 3.14 | 3.22 | 2.92 | 2.44 | 2.19 | 3.52 |
| | Oxygenated Sesquiterpenes | | | 1.72 | 1.31 | 0.96 | 0.76 | 1.00 | 0.72 | 0.66 | 0.68 | 0.43 | 0.52 | 0.40 | 0.98 |
| | Miscellaneous | | | 0.41 | 0.57 | 0.59 | 0.55 | 0.66 | 1.29 | 0.78 | 0.67 | 0.55 | 0.65 | 0.10 | 0.43 |

[a] Compounds listed in order of elution from a BP-20 capillary column; [b] concentrations below 0.05% are marked as tr (traces); [c] total percentage of the 4 main compounds (carvacrol, thymol, γ-terpinene, and *p*-cymene) in the essential oil samples; RIEXP: experimentally determined retention indices; RILIT: retention indices from literature (NIST MS search).

*3.2. Total Phenolic Content and Antioxidant Activity*

As stated in the relevant literature, the primary phenolic constituents found in plants of the Lamiaceae family include phenolic compounds, such as hydroxycinnamic acids, along with flavonoids in the form of esters and glycosides [57,58]. Apart from being influenced by genotype, environmental and handling conditions can also affect the total phenolic content in plants. For this reason, it was essential to determine the actual content of these compounds in the oregano EOs of different geographical origins.

The total phenolic content and antioxidant activity determined for the oregano EO samples are presented in Table 2.

**Table 2.** Mean values and standard deviation (SD) of TPC and antioxidant activity of the oregano sample methanolic extracts for each region studied.

| Location | TPC (mg GAE/g EO) | TEAC (μmol TE/g EO) |
|---|---|---|
| Thessaloniki | 88.25 ± 7.92 | 395.84 ± 12.03 |
| Katerini | 83.20 ± 9.24 | 387.79 ± 13.65 |
| Kilkis | 86.48 ± 5.15 | 382.51 ± 11.37 |
| Kozani | 79.92 ± 7.39 | 457.00 ± 7.42 |
| Volos | 83.92 ± 6.98 | 375.81 ± 9.37 |
| Kalambaka | 85.92 ± 6.41 | 321.89 ± 7.53 |
| Ioannina | 84.17 ± 6.93 | 410.71 ± 10.95 |
| Preveza | 81.58 ± 7.28 | 397.06 ± 6.71 |
| Ileia | 85.38 ± 3.32 | 382.29 ± 20.33 |
| Achaea | 89.03 ± 4.76 | 306.83 ± 5.01 |
| Rethymno | 75.27 ± 3.31 | 461.32 ± 7.27 |
| Heraklion | 74.49 ± 3.57 | 361.43 ± 16.06 |

TPC: total phenolic content; TEAC: Trolox equivalent antioxidant capacity.

All studied samples contained high levels of phenolics without exhibiting any significant fluctuations. The TPC values of all specimens ranged from 74.49 ± 3.57 mg GAE/g EO (Heraklion) to 89.03 ± 4.76 mg GAE/g EO (Achaea). These results were comparable to earlier reports. More specifically, Pasias et al. [59], while investigating the chemical composition of the EOs of aromatic and medicinal herbs cultivated in Greece, reported a TPC value for *Origanum vulgare* L. of 42.6 ± 3.9 mg GAE/g EO. Similarly, Semiz et al. [60], while studying four different *Origanum* species, documented TPC rates ranging from 3.81 to 47.54 mg GAE/g extract. According to Oniga et al. [61], *O. vulgare* ssp. *vulgare* rendered a TPC value of 94.69 ± 4.03 mg GAE/g extract. Moreover, Spiridon et al. [62] compared the TPC values of oregano (*Origanum vulgare*), lavender (*Lavandula angustifolia*), and lemon balm (*Melissa officinalis*) extracts from Romania and found that *O. vulgare* yielded the highest rates of the three, reaching 67.8 ± 3.41 mg GAE/g.

As a general rule, the antioxidant potential of EOs is determined by their chemical composition. Secondary metabolites, such as phenolic compounds, have the ability to bind with double bonds and subsequently exhibit substantial antioxidant activity [63]. Apart from safely preventing food deterioration, natural antioxidants have been reported to help prevent health conditions such as cancer and coronary heart disease [64].

Greek oregano's highly valued antioxidant and antimicrobial activity is strongly associated with the prevalence of the phenols carvacrol and thymol in its essential oil, followed by the abundance of phenolic constituents such as rosmarinic acid and its derivatives within the nonvolatile fraction. A synergistic effect of oxygen-containing compounds has also been proposed [65,66]. Moreover, Al-Mansori et al. [67] investigated potential synergistic effects while examining the antioxidant activity of thymol and carvacrol. It was reported that even though the two phenols exhibited high antioxidant activity, there was no synergistic effect at play.

The Trolox equivalent antioxidant capacity (TEAC) values of the studied samples ranged between 306.83 ± 5.01 μmol TE/g EO (Achaea) and 461.32 ± 7.27 μmol TE/g EO (Rethymno). In their study, Kosakowska et al. [65] found similar results when comparing the antioxidant activity of the EOs and ethanolic extracts of Greek oregano to those of common oregano. Using the DPPH method, the EO of *O. vulgare* ssp. *hirtum* attained a value of 220.29 ± 2.83 μmol Trolox/g EO, whereas the EO of *O. vulgare* ssp. *vulgare* attained a value of 218.78 ± 2.68 μmol Trolox/g EO. Even though one would expect Greek oregano to prevail due to its higher levels of carvacrol and thymol, the amount of oxygenated and hydrocarbon monoterpenes in the common oregano proposedly narrowed the gap. In addition, Rostro-Alanis et al. [68] reported considerable variations while investigating the biological activities of Mexican oregano EOs; the application of the DPPH method produced values ranging from 2.91 to 22,129.54 μmol TE/g EO, depending on the corresponding analyzed fraction.

In tune with the TPC values, no noticeable fluctuations were recorded. No apparent correlation between the antioxidant activity and the total phenolic content of the samples was observed, since high TPC values did not correspond to high TEAC values; the oregano from Achaea attained the highest phenolic content while contrarily yielding the lowest TEAC rates. Simirgiotis et al. [69] also reported a lack of correlation between TPC and TEAC, suggesting that the FC method performed to determine the total phenolic content has certain limitations. Additionally, no association was noted between the chemical composition of the EOs and the TPC or TEAC results. In their study on the antioxidant capacity variation of oregano, Yan et al. [70] reached the same conclusion; no association was established between the oxygen radical absorbance capacity (ORAC) value and EO content of the 352 oregano samples investigated, while only a weak correlation was reported between TPC rates and EO content.

### 3.3. Geographical Differentiation of Greek Oregano Based on EO Composition, TPC, and Antioxidant Capacity

One of the most frequently committed types of fraud in the agricultural market, according to Katerinopoulou et al. [71], is when second-rate agrarian products are promoted as "local". Thus, tools such as the certification of geographic origin must be put into use to safeguard valuable, high-quality products both internationally and within a nation's boundaries.

This study, contrary to that of Vokou et al. [14], investigated the interconnection between the geographical origin of oregano and the composition and properties of its EOs. All 142 samples were initially subjected to MANOVA. However, as several regions overlapped, the samples were divided into two geographical groups: Group A, consisting of Ileia, Heraklion, Kalambaka, Thessaloniki, Kilkis, and Preveza; and Group B, consisting of Rethymno, Volos, Kozani, Katerini, Achaea, and Ioannina.

Sixty oregano samples in Group A and eighty-two samples in Group B were subjected to MANOVA to determine the significant parameters eligible for geographical differentiation. In both groups, dependent variables included essential oil composition, total phenolics, and antioxidant capacity, while geographical origin was set as the independent variable. The significant variables are presented in Table 3.

Thirty-four parameters were detected as significant in Group A ($p < 0.05$) and were further analyzed using LDA. Three statistically significant discriminant functions were generated: Wilks' $\lambda = 0.000$, $X^2 = 661.668$, df = 150 with $p$-value = 0.000 < 0.05 for the first; Wilks' $\lambda = 0.000$, $X^2 = 469.112$, df = 116 with $p$-value = 0.000 < 0.05 for the second; and Wilks' $\lambda = 0.001$, $X^2 = 306.040$, df = 84 with $p$-value = 0.001 < 0.05 for the third, respectively. The variance homogeneity test (Box's M) was insignificant at the 5% significance level (169.516, with F = 1.702, $p$-value = 0.059), indicating the homogeneity of the sample variations for each region. The first discriminant function interpreted 53.5% of the total dispersion with normal distribution $R^2 = 0.995$, the second 25.8% with normal distribution $R^2 = 0.991$, and the third 13.3% with normal distribution $R^2 = 0.982$.

**Table 3.** Statistically significant values in Groups A and B (F-ratio and $p < 0.05$).

| Dependent Variables | Group A | | Group B | |
|---|---|---|---|---|
| | *F* | *Sig.* | *F* | *Sig.* |
| *α*-pinene | 3.747 | 0.006 | 0.707 | 0.620 |
| *α*-thujene | 16.479 | 0.000 | 7.269 | 0.000 |
| camphene | 19.979 | 0.000 | 6.349 | 0.000 |
| *β*-pinene | 43.232 | 0.000 | 12.184 | 0.000 |
| *δ*-3-carene | 19.328 | 0.000 | 7.860 | 0.000 |
| *α*-phellandrene | 16.598 | 0.000 | 16.736 | 0.000 |
| *β*-myrcene | 7.314 | 0.000 | 3.602 | 0.006 |
| *α*-terpinene | 12.833 | 0.000 | 2.341 | 0.050 |
| limonene | 12.281 | 0.000 | 4.912 | 0.001 |
| *β*-phellandrene | 8.004 | 0.000 | 12.216 | 0.000 |
| *γ*-terpinene | 17.102 | 0.000 | 8.733 | 0.000 |
| 3-octanone | 9.956 | 0.000 | 10.266 | 0.000 |
| *p*-cymene | 12.625 | 0.000 | 4.350 | 0.002 |
| *α*-terpinolene | 11.586 | 0.000 | 23.839 | 0.000 |
| 1-octen-3-ol | 13.330 | 0.000 | 10.677 | 0.000 |
| *cis*-sabinene hydrate | 3.399 | 0.010 | 2.573 | 0.033 |
| *trans*-sabinene hydrate | 1.948 | 0.101 | 2.545 | 0.035 |
| bornyl acetate | | | 1.049 | 0.395 |
| caryophyllene | 4.710 | 0.001 | 7.956 | 0.000 |
| 4-terpineol | 14.092 | 0.000 | 1.743 | 0.135 |
| carvacrol methyl ether | 5.480 | 0.000 | 8.447 | 0.000 |
| *cis*-dihydrocarvone | 2.623 | 0.034 | 9.498 | 0.000 |
| *α*-humulene | 8.666 | 0.000 | 16.284 | 0.000 |
| *α*-terpineol | | | 1.253 | 0.293 |
| borneol | 17.003 | 0.000 | 18.737 | 0.000 |
| *β*-bisabolene | 40.163 | 0.000 | 2.540 | 0.035 |
| *δ*-Cadinene | 11.646 | 0.000 | 46.986 | 0.000 |
| *p*-cymen-8-ol | 10.009 | 0.000 | 9.047 | 0.000 |
| carvacryl acetate | | | 2.416 | 0.044 |
| caryophyllene oxide | 20.672 | 0.000 | 13.091 | 0.000 |
| spathulenol | 3.750 | 0.005 | 5.293 | 0.000 |
| 4-isopropyl-m-cresol | 64.496 | 0.000 | 20.953 | 0.000 |
| thymol | 19.504 | 0.000 | 40.470 | 0.000 |
| 5-isopropyl-m-cresol | 28.198 | 0.000 | 22.509 | 0.000 |
| carvacrol | 59.721 | 0.000 | 33.509 | 0.000 |
| TPC | 6.587 | 0.000 | 4.570 | 0.001 |
| TEAC | 2.726 | 0.029 | 26.044 | 0.000 |

The overall interpreted percentage accounted for 92.7% of the total variance, which was highly satisfactory. The values of the group centroids were the average values of the variables as defined by the discriminant functions (Figure 2a). For Ilia, the values were (15.246, −3.748); for Heraklion (−15.280, 1.156); for Kalambaka (−0.247, −8.277); for Thessaloniki (−7.958, 2.675); for Kilkis (1.195, −3.456); and for Preveza (7.853, 13.864). Figure 2a demonstrates the complete separation of the Group A oregano regions.

Thirty-two parameters were found to be significant in Group B ($p < 0.05$) and were further analyzed using LDA. Three statistically significant discriminant functions were generated: Wilks' $\lambda = 0.000$, $X^2 = 671.051$, df = 160 with $p$-value = $0.000 < 0.05$ for the first; Wilks' $\lambda = 0.001$, $X^2 = 448.496$, df = 124 with $p$-value = $0.000 < 0.05$ for the second; and Wilks' $\lambda = 0.009$, $X^2 = 287.835$, df = 90 with $p$-value = $0.001 < 0.05$ for the third, respectively. The variance homogeneity test (Box's M) was insignificant at the 5% significance level (174.858, with F = 1.885, $p$-value = 0.052), indicating the homogeneity of the sample variations of each region. The first discriminant function interpreted 58.5% of the total dispersion with normal distribution $R^2 = 0.987$, the second 20.2% with $R^2 = 0.963$, and the third 12.2% with $R^2 = 0.942$. The overall interpreted percentage accounted for 91.0% of the total variance, which was very satisfactory. The values of the group centroids were the average values of

the parameters. For Achaia, the values were (−0.586, 3.384); for Rethymno (1.464, 7.866); for Volos (−5.917, −1.583); for Katerini (−8.250, −2.060); for Kozani (8.297, −2.618); and for Ioannina (1.457, 0.777). Figure 2b shows the adequate differentiation of the regions of Group B.

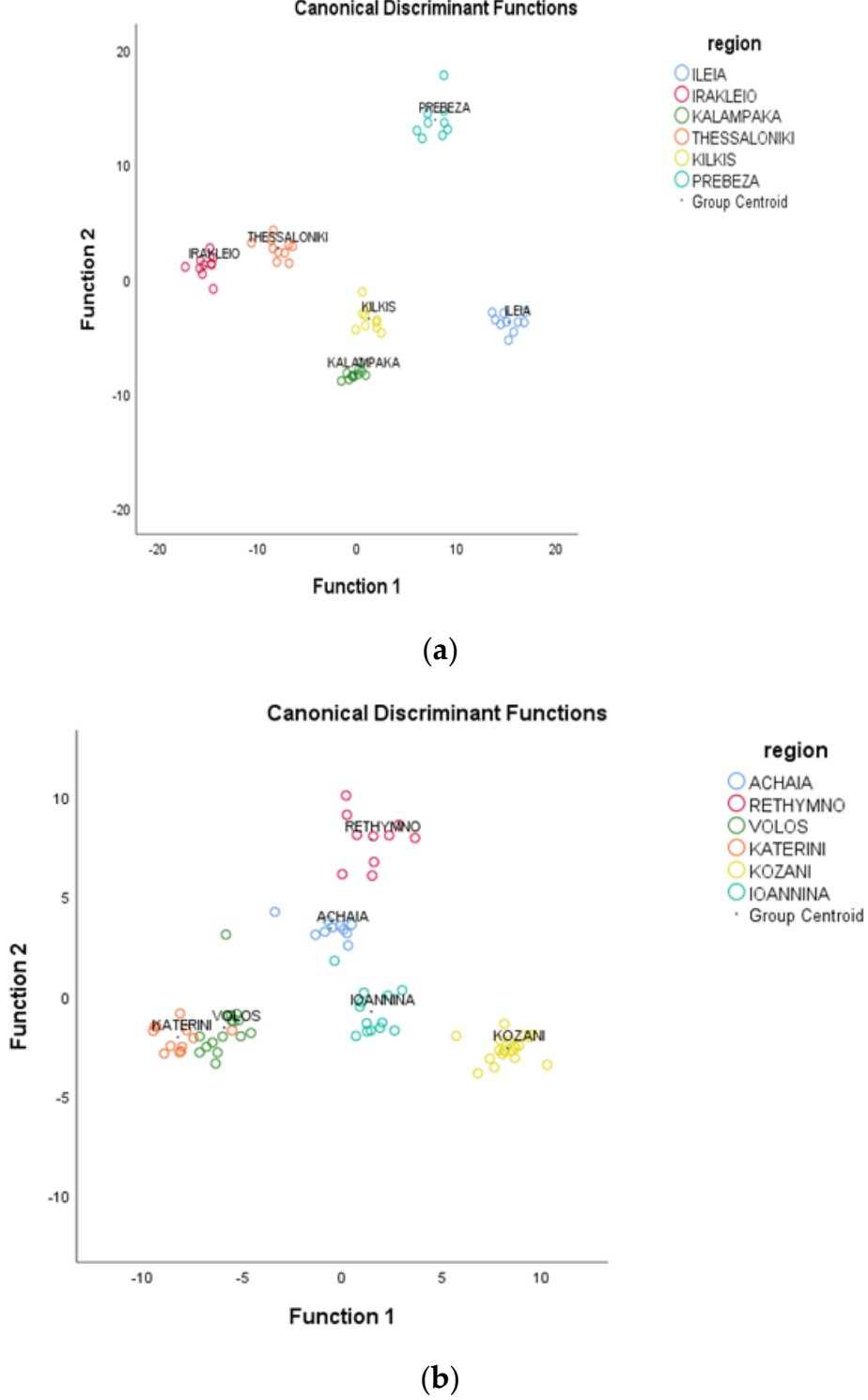

(**a**)

(**b**)

**Figure 2.** (**a**) Group A: oregano geographical differentiation based on essential oil content, total phenolics, and antioxidant capacity. (**b**) Group B: oregano geographical differentiation based on essential oil content, total phenolics, and antioxidant capacity.

In Group A, when using the original method, 100% of all grouped cases were correctly classified; this percentage decreased to 93.3% after the application of the cross-validation method. Regarding Group B, the results of the original method attained a correct classification rate of 98.8% of all grouped cases, while the rate fell to 82.7% with the use of the cross-validation method.

For Groups A and B, the statistical analysis of the EO composition and the TPC and TEAC variables, each separately, provided the results listed in Table 4. The EO composition of Group A presented a very satisfactory separation rate (cross-validation = 93.3%), while the TPC and TEAC values showed inefficient results. Nonetheless, combining the above variables did not seem to affect the results, as the areas' differentiation percentage remained the same after using the cross-validation method. This stability demonstrated the strong effect of the EO composition on the geographical differentiation of oregano. The EO composition of Group B presented a reasonably satisfactory separation rate (cross-validation = 81.5%); nevertheless, the resulting distribution diagram (Figure 2b) seemed to indicate an overlap between the samples of the Rethymno–Achaia and Volos–Katerini areas. Once more, the TPC and TEAC results were unsatisfactory, as shown in Table 4. However, the combination of the above parameters produced good results, and by incorporating more analyses, a better distribution of the samples was attained in the diagram.

**Table 4.** Classification rates of Groups A and B using the original and cross-validation methods (including individual and combinations).

|  | Discriminant Function | Original Method (%) | Cross-Validation (%) |
|---|---|---|---|
| GROUP A | EO compounds | 100 | 93.3 |
|  | TPC and TEAC | 38.3 | 33.3 |
| GROUP B | EO compounds | 96.3% | 81.5% |
|  | TPC and TEAC | 43.9 | 39.0 |

Stepwise LDA was performed as the final step of the statistical data analysis in order to determine the variables with the highest discriminant ability in Groups A and B. Sixteen out of the thirty-four significant EO compounds exhibited a higher discriminant ability in Group A, while in Group B, fourteen out of the thirty-two stood out; in both cases, three statistically significant discriminant functions were formed. As shown in Figure 3, all samples included in Group A were significantly differentiated, while those included in Group B were well differentiated. More specifically, in Group A, the samples from Ileia, Preveza, Heraklion, and Thessaloniki were clearly distinct from all others; however, a small number of those from Kilkis appeared to overlap with those of Kalampaka. On the other hand, in Group B, the specimens originating from Kozani and Achaea were very well differentiated from the rest, contrary to those from Rethymno, which partly overlapped those of Ioannina, and those from Katerini, which overlapped slightly with those from Volos. The overall correct classification rate of Group A was 100% for the original and 100% for the cross-validation method. Respectively, the overall correct classification rate of Group B was 93.9% for the original and 87.8% for the cross-validation method. All in all, 100% correct geographical classification was attained for the regions of Ileia, Preveza, Heraklion, Thessaloniki, and Achaea, followed closely by Kozani (95%) and Rethymno (90%).

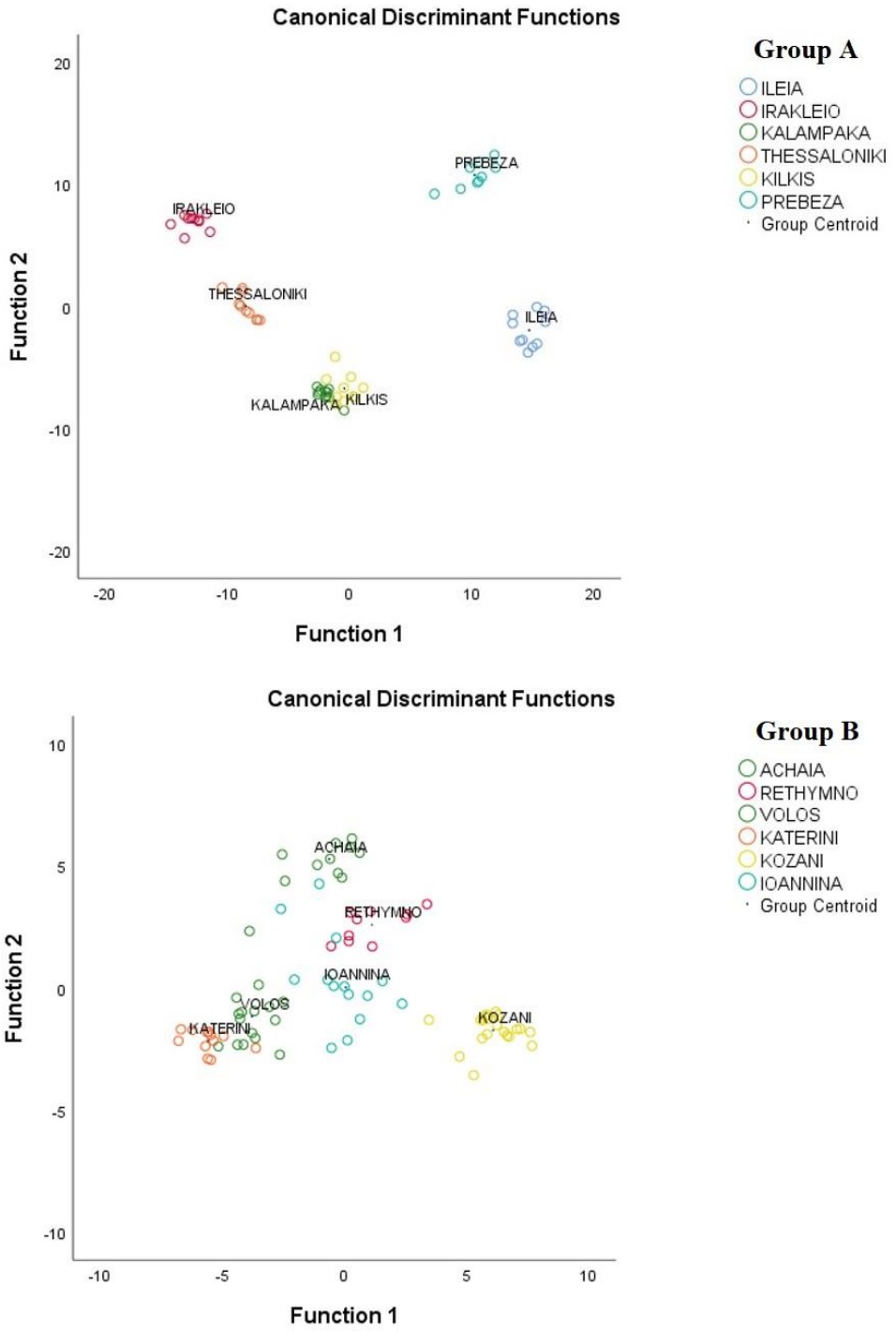

**Figure 3.** Oregano geographical differentiation based on EO composition, total phenolics, and antioxidant capacity. Scatter plot from SLDA analysis (Group A: 100.0% original, 100.0% cross-validation; Group B: 93.9% original, 87.8% cross-validation).

## 4. Conclusions

The chromatographic analysis of Greek oregano EO samples obtained from 12 different locations in Greece indicated remarkable variability in terms of composition, even though the overall content of the four main components (carvacrol, thymol, *p*-cymene, and γ-terpinene) was relatively stable and higher than 85% in all cases.

The statistical treatment of the acquired data yielded satisfactory correct classification rates for both oregano cultivation groups regarding geographical origin. The EO composition was found to be the most significant discriminant parameter (Group A, correct

classification rate 93.3% using the cross-validation method; Group B, correct classification rate 81.5% using the cross-validation method), while TPC and TEAC variables displayed no substantial effect on the geographical differentiation of the samples.

Overall, the acquired results provide preliminary evidence that the chromatographic profile of EOs extracted from Greek oregano samples can act as a powerful tool for geographical origin discrimination purposes, ratifying that GC/MS profiling constitutes an effective approach toward food traceability. Moreover, the implementation of chemometric tools in our study further enabled us to identify which chemical features (markers) are explicitly associated with geographical origin. As the herb market is quite susceptible to fraud and adulteration, a fast and easy methodology such as that described in the current paper could represent a valuable asset for testing authenticity. In addition, apart from their authentication and traceability applications, the techniques used in this work may also act as a tentative guide characterizing the traits and olfactory profiles of commercially available Greek oregano samples originating from different areas of Greece. This guide could be further utilized by the relevant trade authorities when promoting Greek oregano by establishing relative quality standards based on its origin of cultivation.

Nevertheless, the findings of this study must be viewed in light of some limitations. More specifically, the sample size could have been more adequate, and more cultivation regions could have been covered. Even though the results are satisfactory, an even larger sample pool could validate our hypothesis further and eliminate any potential statistical errors. Notwithstanding the preliminary nature of our research, the methods considered in this work represent a promising solution for Greek oregano traceability, thus deserving further investigation.

**Author Contributions:** Conceptualization, A.V.B.; data curation, E.S.T. and I.S.K.; investigation, E.S.T.; formal analysis, E.S.T.; methodology, A.V.B., I.S.K. and E.S.T.; supervision, A.V.B.; writing—original draft, E.S.T. and I.S.K.; writing—review and editing, A.V.B., I.S.K. and E.S.T.; project administration, A.V.B. All authors have read and agreed to the published version of the manuscript.

**Funding:** This research received no external funding.

**Institutional Review Board Statement:** Not applicable.

**Data Availability Statement:** Not applicable.

**Acknowledgments:** We sincerely thank all the growers who provided us with the free oregano samples used in this study.

**Conflicts of Interest:** The authors declare no conflict of interest.

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
