# Peer review of "Chemometric Screening of Oregano Essential Oil Composition and Properties for the Identification of Specific Markers for Geographical Differentiation of Cultivated Greek Oregano"

_sustainability, doi:10.3390/su142214762_

Round 1

Reviewer 1 Report

General comments:

Dear authors, it seems to me that this manuscript has great relevance in the scientific world. However, many points affect the quality of the manuscript.

My biggest concern is the low number of experimental units and the sampling methods. You need to be more specific, because the reasons for seeing variations in the EO content of oregano are many. What about the soil characteristics? The characteristics of the climate? The use or not of chemical or organic fertilizers? The age of the plant, etc. Perhaps your results are not influenced by the region and but by the different forms of cultivation.

Abstract: The abstract says a lot about the multivariate analysis and about the samples, but what were your main results and your conclusion? The conclusion described here does not specifically correlate with the title.

Introduction: The introduction is very generic, add as many numbers as you can (Eg carvacrol content ranges from 10 to 70 ppm). The introduction describes the essential oil content of oregano and a few lines on chemometric analysis. However, according to the objective, chemometric analysis is the focus; so why not add more information on the use of Chemometric Analysis in similar or analogous studies. Add the hypothesis of the manuscript.

Material and methods: In the tables I can see the analysis of the Duncan test, the effect of Neem leaves, the effect of PEG and the p-value of the Neem leaves*PEG interaction. But where are the tables or figures that explain the decomposition of these interactions?

Discussion: The discussion is very generic, with many beliefs and scientific data to support its results, which are ultimately just a scientific review. Comparing your results with other authors is fine; however, the discussion topic is to explain biologically how you obtained these results. What were the biological mechanisms? How much was changed (gr, %, etc.)? Etc.

The conclusion must be objective and direct. This way is a comment of the authors. Also I don't see a clear conclusion because this conclusion is not correlated with the objective.

Specific comments:

Line 85: Why do you use this low number of samples? I think that in two years many more samples could have been collected. In addition, considering the variability of the chemical composition and the EO, perhaps a low number of samples could promote results affected by the statistical type II error.

Line 89: Why stems? What is the EO content in the leaves? Perhaps the leaves have higher EO content.

Line 94: What does “HD” mean? Describe the name in full the first time it appears in the manuscript considering the abstract and the body of the manuscript as different documents.

Lines 101-103: What do “GC, MS and MSD” mean?

Line 122: What does "TCP" mean?

Line 140: What does “DPPH” mean?

Lines 155 and 160: What do “LDA and SLDA” mean?

Line 200: “align with the findings of past studies” … What was the range found in these studies?  

Lines 200-201: “speak highly of the excellent quality of the acquired samples” … Please, be more specific.

Lines 204-212: This is the issue of results; in this way these lines belong to the topic of discussion.

Lines 214-215: Delete these lines because these are not relevant.

Lines 216-221: I'm so sorry, I'm lost. Were the results and discussion topics presented in a single topic? The author's instruction allows these themes to be combined; however, if possible, separate those topics.

Lines 224-225: “These … reports.” This idea is irrelevant. Remove it.

Lines 241-242 and line 288: Delete these lines because these are not relevant.

Figure 2a: What do "Function 1 and 2" mean?

Lines 332-333: Delete these lines because these are not relevant.

Figure 2a: What does "poor" mean? It is not a formal word.

Lines 363-364: Irrelevant. Remove it.

Author Response

Response to comments

Reviewer 1:

Dear authors, it seems to me that this manuscript has great relevance in the scientific world. However, many points affect the quality of the manuscript.

Comment: My biggest concern is the low number of experimental units and the sampling methods. You need to be more specific, because the reasons for seeing variations in the EO content of oregano are many. What about the soil characteristics? The characteristics of the climate? The use or not of chemical or organic fertilizers? The age of the plant, etc. Perhaps your results are not influenced by the region and but by the different forms of cultivation.

Response: We have considered your notes and decided to add potential limitations to our study concerning the number of samples. As you can see in the revised text l. 119-122, we have also included some details regarding cultivation practices. However, as the geographical characteristics of a region include climate, soil, and hydrology, we believe our geographical differentiation was valid without having to study them separately.

 Comment: Abstract: The abstract says a lot about the multivariate analysis and about the samples, but what were your main results and your conclusion? The conclusion described here does not specifically correlate with the title.

Response: We have made the appropriate changes in order to address this issue. See revised text l. 17-22.

Comment: Introduction: The introduction is very generic, add as many numbers as you can (Eg carvacrol content ranges from 10 to 70 ppm). The introduction describes the essential oil content of oregano and a few lines on chemometric analysis. However, according to the objective, chemometric analysis is the focus; so why not add more information on the use of Chemometric Analysis in similar or analogous studies. Add the hypothesis of the manuscript.

Response: We have added the information the reviewer suggested. See revised text l. 56-58, 67-90.

Comment: Material and methods: In the tables I can see the analysis of the Duncan test, the effect of Neem leaves, the effect of PEG and the p-value of the Neem leaves*PEG interaction. But where are the tables or figures that explain the decomposition of these interactions?

Response: We are confused. Our manuscript is dealing with oregano, not neem leaves.

 Comment: Discussion: The discussion is very generic, with many beliefs and scientific data to support its results, which are ultimately just a scientific review. Comparing your results with other authors is fine; however, the discussion topic is to explain biologically how you obtained these results. What were the biological mechanisms? How much was changed (gr, %, etc.)? Etc.

Respose: Our study deals with a geographical differentiation of cultivated Greek Oregano using chemimetrics. If we were to include any biological mechanisms we could get out off topic and confound the reader.

Comment: The conclusion must be objective and direct. This way is a comment of the authors. Also, I don't see a clear conclusion because this conclusion is not correlated with the objective.

Response: We have made the necessary changes to address this issue. See revised text Conclusion section.

 Specific comments:

 Comment:

Line 85: Why do you use this low number of samples? I think that in two years many more samples could have been collected. In addition, considering the variability of the chemical composition and the EO, perhaps a low number of samples could promote results affected by the statistical type II error.

Response: Our samples had to be of similar cultivation principles (organic, open filed etc.) so we did not have a lot of options to choose from. See revised text l.119-122.

We understand reviewer’s concerns and we have included them in our conclusions as limitations to our study. See revised text l. 400-403.

Comment: Line 89: Why stems? What is the EO content in the leaves? Perhaps the leaves have higher EO content.

Response: By “stems” we meant the aerial part of the plant. We used the word “stems” because we wanted to make clear that all oregano samples were treated/rubbed by us to retain homogeneity. However, we have addressed the issue by replacing the word “stems” with “aerial parts”. See revised text l. 116.

 Comment: Line 94: What does “HD” mean? Describe the name in full the first time it appears in the manuscript considering the abstract and the body of the manuscript as different documents.

Response: Noted, all abbreviations have been dealt with. See throughout the revised text.

Comment: Lines 101-103: What do “GC, MS and MSD” mean?

ResponseNoted, all abbreviations have been dealt with. See throughout the revised text.

Comment: Line 122: What does "TCP" mean?

Response: Noted, all abbreviations have been dealt with. See throughout the revised text.

Comment: Line 140: What does “DPPH” mean?

Response: Noted, all abbreviations have been dealt with. See throughout the revised text.

Comment: Lines 155 and 160: What do “LDA and SLDA” mean?

Response: Noted, all abbreviations have been dealt with. See throughout the revised text.

Comment: Line 200: “align with the findings of past studies” … What was the range found in these studies? 

Response: We added the requested information. See revised text l. 231-233.

Comment: Lines 200-201: “speak highly of the excellent quality of the acquired samples” … Please, be more specific.

Response: We have made the necessary changes. See revised text l. 231-233.

Comment: Lines 204-212: This is the issue of results; in this way these lines belong to the topic of discussion.

Response: The reviewer is right. Results & Discussions are registered together.

Comment: Lines 214-215: Delete these lines because these are not relevant. 

Response: These lines (revised text l. 242-243) point to Table 2 where the results are demonstrated. Why does the reviewer consider them inrelecant?  

Comment: Lines 216-221: I'm so sorry, I'm lost. Were the results and discussion topics presented in a single topic? The author's instruction allows these themes to be combined; however, if possible, separate those topics.

Response: The reviewer is right. Results & Discussions are registered together 

Comment: Lines 224-225: “These … reports.” This idea is irrelevant. Remove it.

Response: We have rephrased the text. Revised text l. 246-247.

Comment: Lines 241-242 and line 288: Delete these lines because these are not relevant.

Response: We have altered the text. See revised text l. 264. Regarding l. 288 did the reviewer meam l. 248. If so we have altered the text. See revised text l. 270.

Comment: Figure 2a: What do "Function 1 and 2" mean?

Response: In the original method, the prediction rate results from the contribution of all cases in the discriminant functions while in cross-validation, a randomly chosen parameter, is classified in a group based on a discriminant function, created by all the other parameters (except the randomly chosen one). This procedure is repeated every time for all the parameters of the tested sample. Software (SPSS) generated two-dimension figures correspond to the original method as it is extremely difficult to generate a single figure from the cross-validation method. Function 1-horizontal axes highly differentiate Preveza, Ileia, Kilkis, and Kalampaka, while function 2-vertical axes differentiate Thessaloniki and Irakleio, but this differentiation is not as high as in function 1. The average values of the variables that contribute to the differentiation as defined by the discriminant functions are given by group centroids. The same stands for every figure given.

Comment: Lines 332-333: Delete these lines because these are not relevant.

Response: We have deleted them as suggested.

Comment: Figure 2a: What does "poor" mean? It is not a formal word.

Response: Figure 2a does not include the word “poor”. Nonetheless, poor (meaning unsatisfactory) is officially used to describe results, performance, or even yield. We can provide documentation if requested.

 Comment: Lines 363-364: Irrelevant. Remove it.

Response: We have deleted them as suggested.

Reviewer 2 Report

The study reports the EO compositions, total phenolic contents,  and antimicrobial activities of 142 Origanum vulgare  subsp.  hirtum samples from different regions of Greece. A similar study undertaken with multiple samples collected from the wild in the northwestern regions of Turkey has resulted in the registration of Baser (Carvacrol-rich) and Tinmaz (Thymol-rich) cultivars with the Ministry of Food, Agriculture and Animal Husbandry of Turkey in 2017. The authors do not indicate the intention of this study clearly was it done for a selection study or not. Several similar studies done in Turkey and published in peer-reviewed journals are not included in the references. 

No herbarium voucher information for the plant materials is given.

GC/MS data are  good for qualitative purposes but for quantitative assessments of the detected  components GC-FID should have been resorted to during the analysis for more consistent results. 

In Table 1, symbols of compound names are given in capital letters. This is wrong and confusing. They must be given with their respective symbol like  in Table 3.

Since during the GC/MS analysis only a normal-phase polar column was used, enantiomers cannot be separated. However, in the list of compounds (+)-spathulenol and dl-limonene are cited. Chiral columns only can separate enantiomers. d-cadinene (probably delta-cadinene) must be given with its appropriate delta symbol. 

para-thymol and meta-thymol are not accepted names of natural thymol derivatives. Their formulae must be written clearly. 

Author Response

Response to comments.

Reviewer 2:

Comment: The study reports the EO compositions, total phenolic contents, and antimicrobial activities of 142 Origanum vulgare subsp.  hirtum samples from different regions of Greece. A similar study undertaken with multiple samples collected from the wild in the northwestern regions of Turkey has resulted in the registration of Baser (Carvacrol-rich) and Tinmaz (Thymol-rich) cultivars with the Ministry of Food, Agriculture and Animal Husbandry of Turkey in 2017. The authors do not indicate the intention of this study clearly was it done for a selection study or not. Several similar studies done in Turkey and published in peer-reviewed journals are not included in the references. 

Response: We were unable to find the specific study the reviewer proposed, however, we have updated our references to the suggestions. See the updated ref. list in the revised text.

Comment: No herbarium voucher information for the plant materials is given.

Response: The oregano plants used in this study originated from cultivations, not the wild. We have added information regarding the origin of the certified samples. See revised text l. 119-122.

Comment: GC/MS data are good for qualitative purposes but for quantitative assessments of the detected components GC-FID should have been resorted to during the analysis for more consistent results. 

Response: We have made a qualification and semi-quantification of the volatiles using an internal standard and not a strict quantification. Our goal was to give a trend of the volatile content. The GC/MS could be used for quantification purposes using the SIM (single ion monitoring) mode and to give consistent and comparable results with the GC-FID. But, in both cases we need standard compounds for the construction of calibration curves of selected volatile compounds.   

Comment: In Table 1, symbols of compound names are given in capital letters. This is wrong and confusing. They must be given with their respective symbol like in Table 3.

Response: The issue has been addressed. See revised text.

Comment: Since during the GC/MS analysis only a normal-phase polar column was used, enantiomers cannot be separated. However, in the list of compounds (+)-spathulenol and dl-limonene are cited. Chiral columns only can separate enantiomers. d-cadinene (probably delta-cadinene) must be given with its appropriate delta symbol. 

Response: We presented the compounds according to the suggestions of the Wiley Library database and cross-referenced them with the RI indices, however, we have made the appropriate changes.

Comment: para-thymol and meta-thymol are not accepted names of natural thymol derivatives. Their formulae must be written clearly. 

Response: We have made the suggested alterations. See throughout revised text.

Reviewer 3 Report

I consider that the work is well written, it is well understood, it needs some minor corrections in the text.

line 37:  "Origanum vulgare",  I think it's in italics.

lines 45, 55, 179, 196, 209, 210 and 366 : p-cymene: is not with "p" it's a symbol

line 179: trans- is with italic

Table 1. please correct the names of the compounds with the symbols to be used

Table 3.  name symbols are in italics, what is "F"?

Author Response

Response to comments

Reviewer 3:

I consider that the work is well written, it is well understood, it needs some minor corrections in the text.

Comment: line 37:  "Origanum vulgare”, I think it's in italics.

Response: We have addressed the issue. See throughout the revised text.

Comment: lines 45, 55, 179, 196, 209, 210 and 366: p-cymene: is not with "p" it's a symbol

Response: We have made all the necessary changes. See throughout the revised text.

Comment: line 179: trans- is with italic

Response: We have made all the necessary changes. See throughout the revised text.

Comment: Table 1. please correct the names of the compounds with the symbols to be used

Response: We have made all the necessary changes. See the revised text.

Comment: Table 3.  name symbols are in italics, what is "F"?

Response: it is a statistically generated value that indicates the ratio of two mean square values. If the null hypothesis is true, we expect F to have a value close to 1.0 most of the time. A large F-ratio means that the variation among group means is more than we would expect to see by chance.

Reviewer 4 Report

This manuscript entitled “Chemometric Screening of Oregano Essential Oil Composition and Properties for the Identification of Specific Markers for Geographical Differentiation of cultivated Greek Oregano” by Tsoumani et al. deals with the analysis of Greek oregano collected from different geographical areas of Greece. After a careful reading, I found this manuscript appropriate and suitable for publication in the Sustainability journal. However, I have pointed out several grey points in the manuscript which need specific attention. Thus, the manuscript can be recommended for acceptance pending suitable major revisions. My specific comments are:

1.      The language of the manuscript is messy and most of the sentences are written in passive form. I suggest improving the writing style in terms of fluency and logical order.

2.      The abstract should indicate the major problem followed by objectives, methods, major numerical results, major outcome, and significance of the study in a single paragraph. The current version of the abstract lacks a proper logical flow of reading.

3.      The introduction is fine but requires some improvements in the writing style of botanical names and their authorities. Most of them are not italicized and lack authority names.

4.      Line 73: remove “project” and add “study”.

5.      Extend the figure and table captions to become more informative. Currently, they are very ordinary and give no clear information.

6.      Never use numerical readings, author names, and abbreviations while starting a paragraph (Line 108).

7.      Given description of some statistical parameters is not necessary. It is better to remove them and only provide relevant references (Line 160-173).

8.      Be consistent while writing compound/column names in Table 1. Do not use UPPERCASE.

9.      Define TPC/TEAC abbreviations under Table 2-footer.

10.   Results are clearly presented as text, but their comparison and logical interpretation using recent studies are lacking.

11.   Do not use discussion under conclusion. This section should include major findings, shortcomings, and future recommendations only.

12.   References: should be updated to the latest ones.

Author Response

Response to comments

Reviewer 4:

This manuscript entitled “Chemometric Screening of Oregano Essential Oil Composition and Properties for the Identification of Specific Markers for Geographical Differentiation of cultivated Greek Oregano” by Tsoumani et al. deals with the analysis of Greek oregano collected from different geographical areas of Greece. After a careful reading, I found this manuscript appropriate and suitable for publication in the Sustainability journal. However, I have pointed out several grey points in the manuscript which need specific attention. Thus, the manuscript can be recommended for acceptance pending suitable major revisions. My specific comments are:

Comment: The language of the manuscript is messy and most of the sentences are written in passive form. I suggest improving the writing style in terms of fluency and logical order.

Answer: We have made some changes throughout the manuscript. In addition, a native speaker proofread our paper, and we followed his suggestions concerning the appropriate writing style.

Comment:  The abstract should indicate the major problem followed by objectives, methods, major numerical results, major outcome, and significance of the study in a single paragraph. The current version of the abstract lacks a proper logical flow of reading.

Answer: We have made all the necessary changes. See revised text l. 17-22.

Comment: The introduction is fine but requires some improvements in the writing style of botanical names and their authorities. Most of them are not italicized and lack authority names.

Answer: We have made all the necessary changes. See changes throughout the revised text.

Comment: Line 73: remove “project” and add “study”.

Answer: We have made all the necessary changes. See revised text l. 102.

Comment: Extend the figure and table captions to become more informative. Currently, they are very ordinary and give no clear information.

Answer: We have made the appropriate changes according to reviewer’s suggestions. More explanations regarding the figures and tables are given in the Results and Discussion section in the revised text.

Comment: Never use numerical readings, author names, and abbreviations while starting a paragraph (Line 108).

Answer: We have made all the necessary changes. See changes throughout the revised text.

Comment: Given description of some statistical parameters is not necessary. It is better to remove them and only provide relevant references (Line 160-173).

Answer: We have made all the necessary changes. See revised text l. 196-199.

Comment: Be consistent while writing compound/column names in Table 1. Do not use UPPERCASE.

Answer: We have made all the necessary changes. See revised Table 1.

Comment: Define TPC/TEAC abbreviations under Table 2-footer.

Answer: We have made all the necessary changes. See revised Table 2.

Comment: Results are clearly presented as text, but their comparison and logical interpretation using recent studies are lacking.

Answer: We have tried to resolve the issue by using updated studies.

Comment: Do not use discussion under conclusion. This section should include major findings, shortcomings, and future recommendations only.

Answer: We have made several alterations as suggested. See revised conclusion section.

Comment: References: should be updated to the latest ones.

Answer: We have followed reviewer’s suggestions accordingly, wherever possible. See updated ref. list in the revised text.

Round 2

Reviewer 1 Report

Dear authors, I agree with the changes made.

Author Response

Thank you very much

Reviewer 2 Report

There are minor editorial corrections as follows:

line 419, 505 and 531 Origanum vulgare ssp. hirtum

line 531 O. onites

line 130 - the ref. 39. Esen, G.; Azaz, A.D.; Kurkcuoglu, M.; Baser, K.H.C.; Tinmaz, A. Essential Oil and Antimicrobial Activity of Wild and Culti- 504 vated Origanum vulgare L. subsp. hirtum (Link) Letswaart from the Marmara Region, Turkey. Flavour Fragr. J. 2007, 22, 371–376, 505 doi:10.1002/ffj.1808. could have been included in the discussion. Its citation in the text is quite irrelevant. 

Author Response

Comment: line 419, 505 and 531 Origanum vulgare ssp. hirtum

Response: The reviewer is right. We have made the appropriate changes. See revised text l. 422, 509, 535.

Comment: line 531 O. onites

Response: The reviewer is right. We have made the appropriate changes. See revised text l. 536.

Comment: line 130 - the ref. 39. Esen, G.; Azaz, A.D.; Kurkcuoglu, M.; Baser, K.H.C.; Tinmaz, A. Essential Oil and Antimicrobial Activity of Wild and Culti- 504 vated Origanum vulgare L. subsp. hirtum (Link) Letswaart from the Marmara Region, Turkey. Flavour Fragr. J. 2007, 22, 371–376, 505 doi:10.1002/ffj.1808. could have been included in the discussion. Its citation in the text is quite irrelevant.

Response: The reviewer is right. We wanted to include this study in the discussion, but we were pressed for time, that is why we chose this way. We have made the suggested alterations. See revised text l.224-226

Reviewer 4 Report

All major concerns and comments are implemented in the revised manuscript.  The manuscript can be accepted in current form.

Author Response

Thank you very much.